# Mass Casualty Decontamination for Chemical Incidents: Research Outcomes and Future Priorities

**DOI:** 10.3390/ijerph18063079

**Published:** 2021-03-17

**Authors:** Samuel Collins, Thomas James, Holly Carter, Charles Symons, Felicity Southworth, Kerry Foxall, Tim Marczylo, Richard Amlôt

**Affiliations:** 1Chemicals and Environmental Effects Department, Centre for Radiation, Chemical and Environmental Hazards, Public Health England, Didcot OX11 0RQ, UK; tom.james@phe.gov.uk; 2COVID-19 Behavioural Science and Insights Unit, Public Health England, Public Health England, London SE1 8UG, UK; holly.carter@phe.gov.uk (H.C.); charles.symons@phe.gov.uk (C.S.); richard.amlot@phe.gov.uk (R.A.); 3Behavioural Science, Emergency Response Department Science & Technology, Health Protection Directorate, Public Health England, Porton SP4 0JG, UK; felicity.southworth@phe.gov.uk; 4Toxicology Department, Centre for Radiation, Chemical and Environmental Hazards, Public Health England, Didcot OX11 0RQ, UK; Kerry.foxall@phe.gov.uk (K.F.); tim.marczylo@phe.gov.uk (T.M.); 5Department of Psychological Medicine, Institute of Psychiatry, Psychology and Neuroscience, King’s College, London SE1 1UL, UK

**Keywords:** decontamination, chemical, mass casualty

## Abstract

Planning for major incidents involving the release of hazardous chemicals has been informed by a multi-disciplinary research agenda which has sought to inform all aspects of emergency response, but with a focus in recent years on mass casualty decontamination. In vitro and human volunteer studies have established the relative effectiveness of different decontamination protocols for a range of chemical agents. In parallel, a programme of research has focused on communicating with and managing large numbers of contaminated casualties at the scene of an incident. We present an accessible overview of the evidence underpinning current casualty decontamination strategies. We highlight where research outcomes can directly inform response planning, including the critical importance of beginning the decontamination process as soon as possible, the benefits of early removal of contaminated clothing, the evidence under-pinning dry and wet decontamination and how effective communication is essential to any decontamination response. We identify a range of priority areas for future research including establishing the significance of the ‘wash-in’ effect and developing effective strategies for the decontamination of hair. We also highlight several areas of future methodological development, such as the need for novel chemical simulants. Whilst considerable progress has been made towards incorporating research outcomes into operational policy and practice, we outline how this developing evidence-base might be used to inform future iterations of mass casualty decontamination guidance.

## 1. Introduction

The global development, distribution and use of chemicals continues to increase annually, with thousands, including toxic industrial chemicals, being manufactured and distributed in excess of one metric tonne annually [1]. Though all chemicals manufactured above one tonne are registered, which requires a fundamental level of toxicological assessment for many chemicals in current use, an in-depth assessment of hazard has not been completed. Accompanying these developments is an increasing risk of a chemical incidents leading to human exposure to and injury from hazardous chemicals. A chemical incident is defined as “an unexpected, uncontrolled release of a chemical from its containment” [2]. Chemical incidents may be caused by accidental (e.g., chemical spillages, fires) or deliberate (e.g., terrorist) factors. Globally, there are multiple chemical incidents involving the exposure, or potential exposure of tens of thousands of people every year [1,3]. Recent accidents including the 2013 Lac-Mégantic rail disaster [4] and the 2015 Tianjin explosions [5] accompanied by an alarming rise in the criminal dumping of chemical waste [6] and deliberate use of chemical agents including acid attacks [7], Sarin in Syria [8], VX in Malaysia [9] and Novichok in the United Kingdom (UK) and Russia [10] all highlight the ever-present threat from chemical incidents.

During a chemical incident people may be exposed to chemicals by several routes including by inhalation or deposition of an airborne chemical on skin, hair and eyes or through direct contact with a liquid or solid contaminant [1,11]. The primary public health concern following a chemical incident will be to preserve life. To this end, the decontamination of exposed persons, defined as any action that reduces, removes, neutralizes or inactivates contamination is an immediate consideration. Decontamination not only limits the potential exposure (e.g., reducing contaminant absorption through the skin) but also reduces the uncontrolled spread of contamination to unexposed persons, emergency responders, equipment and health care facilities, a scenario highlighted by the 1995 Tokyo subway sarin attacks [12].

Occasionally, chemical incidents may affect large numbers of people. These so-called ‘mass-casualty’ incidents are defined by NHS England as ‘an incident (or series of incidents) causing casualties on a scale that is beyond the normal resources of the emergency and healthcare services’ ability to manage’ [13]. Such incidents are complex and first responders face challenges in implementing appropriate and structured decontamination including the volume of casualties, exposed persons leaving the scene, delays to medical intervention and access to necessary specialist resources [14,15,16,17]. It is vital that decontamination strategies are underpinned by robust scientific evidence to ensure that decontamination is well-planned, timely and efficacious. A substantial body of research, led predominantly by researchers in the UK and USA, has emerged resulting in substantive changes to the planning and implementation of mass-casualty decontamination for chemical incidents in recent years.

Following chemical incidents in the UK, first responders have historically been instructed to stand-off and wait for specialist Fire and Rescue Services (FRS) capability including mass-decontamination units (MDUs) to arrive at the scene. However, this approach introduces a time delay that is especially detrimental to persons exposed to rapidly acting hazardous chemicals. Consequently, informed by the research outputs discussed in this paper (including the Public Health England (PHE)-led ORCHIDS programme [18,19,20,21,22]), UK operational response moved from a reliance on Specialist Operational Response (SOR), towards rapidly deployable strategies such as emergency evacuation, disrobe and decontamination using improvised dry, improvised wet (rinse-wipe-rinse [RWR]) and interim wet methods, known collectively as the Initial Operational Response (IOR) [23]. The purpose of IOR is to rapidly decontaminate casualties shortly after exposure. The UK Joint Emergency Services Interoperability Programme (JESIP) has produced a protocol; “STEPS 1-2-3 Plus” to guide non-specialist emergency responders to undertake these life-saving actions in the event of a chemical, biological, radiological or nuclear (CBRN) incident [23,24].

An important feature of IOR is the emphasis placed on clear and effective communication from responders to affected casualties, to build trust and ensure that decontamination procedures are timely and structured. Informed by a Joint Dynamic Hazard Assessment that considers multiple factors about the incident and the requirements of all emergency response organisations present. IOR can be succeeded by a Specialist Operational Response (SOR) in which MDUs are used to decontaminate up to hundreds of exposed persons per hour.

In the US, prior to 2015, whilst some national guidance existed, planning was conducted at the state level. National guidance was military in origin and was based on ‘perceived best practice’. Federal Primary Response Incident Scene Management (PRISM) guidance introduced in 2015 [25] is based on a new programme of research and usefully summarised key new evidence in a range of domains. Similar to UK IOR guidance [23], the PRISM approach focuses on the initial “disrobe and decontaminate” response to chemical incidents, placing strong emphasis on the speed of the initial response to ensure casualty survival and reduce morbidity. The second edition of PRISM published in 2019 [26] extended and developed these approaches in key areas, including a recommended triple decontamination protocol, a combination of dry, ‘Ladder Pipe’ (interim) and technical (mass) decontamination, the decontamination of hair and scene decision-making.

A review by Monteith and Pearce [27] summarises previous concepts for self-care decontamination following a mass casualty chemical incident. This review synthesises recent key findings from published in vitro and human volunteer studies and describes how they have informed current IOR and PRISM strategies for the UK and the US. Despite significant advances, we discuss considerable knowledge gaps that should be prioritized as future research needs.

## 2. Key Findings from Mass Casualty Decontamination Research

Mass casualty chemical decontamination protocols have been systematically developed, optimised and evaluated through a series of in vitro and human volunteer trials. The evidence generated during these studies has improved our understanding of several critical areas of the decontamination process, ultimately leading to enhanced decontamination capabilities. Major research findings underpinning present-day UK and US decontamination approaches are discussed below.

### 2.1. Speed Is Critical

Until 2006, the UK ‘Model Response’ for decontamination comprised of two distinct phases. Upon first responder arrival at a scene of unknown aetiology, guidance known as “STEP 1-2-3” was followed. If three or more casualties were affected the instruction was for first responders to stand off (phase one) and wait for specialist resources to arrive (phase two). The arrival time of specialist resources in the UK may be up to one hour. The same was also true for the US approach pre-2015. A time delay of >20 min was predicted before the arrival of fire services to conduct gross wet decontamination using a ‘Ladder Pipe System’ [28]. These delays in initiation of decontamination and casualty management could have significant detrimental effects on casualty morbidity and mortality especially for chemical agents that act via rapid dermal absorption or inhalation, as well as knock on effects on receiving healthcare facilities who may see a large number of self-presenting, potentially contaminated casualties.

Following a review of the UK model response, underpinned by conclusions from the Public Health England-led ORCHIDS decontamination projects [22], it was agreed that a more rapid intervention was needed to minimise the risk of morbidity and mortality. The updated model was named the Initial Operational Response (IOR) and is summarised as a new “STEP 1-2-3 Plus” protocol. In the event of three or more casualties, emergency personnel now assume a HAZMAT/CBRN incident and immediately start IOR: the removal of casualties from the scene or site of gross contamination, disrobing of casualties to remove potentially contaminated clothing, and improvised dry or wet decontamination with any readily available absorbent materials and water source as necessary. Following improvised decontamination, interim decontamination would be conducted, using fire service vehicles to create a shower corridor through which casualties are showered using hoses in a more controlled manner (also known as the ‘Ladder Pipe’ system in the US). These initial and interim steps significantly reduce the time between exposure and intervention.

Informed by the same evidence and studies funded by the Biomedical Advanced Research and Development Authority (BARDA) [26] the US has also moved to an approach focusing on rapid intervention including evacuation, disrobe and improvised decontamination. Both approaches recognise that while IOR and interim procedures may not be as effective as SOR, rapid chemical removal is preferable to waiting and doing nothing.

### 2.2. The Importance of Disrobing

Removal of clothing following chemical exposure is widely recognised as an appropriate early intervention to prevent both the penetration of the chemical to the casualties’ skin and its transfer to others including emergency staff. Although this approach makes intuitive sense and 80–90% removal of chemicals by disrobing has been suggested [26], there is limited published evidence to demonstrate this efficacy [29]. There are two competing aspects to consider with clothing during a chemical incident. Clothing initially acts as a barrier preventing contact with skin but also acts as a source of exposure where volatile agents are released from clothing contributing to inhalation exposure. The rate of off gassing from clothing depends upon the type of material. For example, a study using the mustard simulant methyl salicylate demonstrated a range of off-gassing times for contaminated clothing from 7 min for denim up to 42 min for a down jacket [30]. In vitro, dermatomed skin studies show that clothing significantly decreases penetration of a selection of chemical warfare agents but that these effects are temporal [31]. Once a chemical has penetrated to skin, clothing may increase the absorption across the skin as has been demonstrated for chlorine [32]. However, it is not established whether these effects would be observed under realistic conditions.

Most replies to a US survey of key responders said they would initiate disrobe immediately whereas others replied that their response would be situation dependent [33]. Response may be dependent upon the degree of contamination, the identity of the chemical agent or considerations such as the ambient temperature with a potential for hypothermia. ‘Disrobe’ packs (containing temporary replacement clothing) where available provide little protection from hypothermia, however they are a useful solution to modesty concerns of exposed public. In the UK and the US, emergency disrobe and decontamination forms part of the IOR. Disrobing is recommended within 15 min of exposure and responders should consider hypothermia and modesty concerns by using alternative clothing or blankets where available [23,26]. This guidance also highlights the importance of retaining spent clothing for evidence purposes and the need to avoid transfer of chemical agent to skin from clothing. The latter can be achieved by wearing gloves during disrobe (provided in UK disrobe packs) and by cutting off clothing with scissors (also in UK disrobe packs) in preference to removing clothes over the head. The use of scissors to cut off clothing has been shown to be particularly useful for non-ambulant casualties [34].

### 2.3. Dry Decontamination of Skin Is an Effective Intervention for Liquids When Initiated in a Timely Manner

Most decontamination strategies involve the use of water (with or without detergents or bleach) to rinse and remove chemicals from skin and hair [35]. Whilst water is a readily available commodity in developed countries, establishing a structured wet decontamination process for mass-casualty incidents involves the deployment of specialist resources including front-line Fire and Rescue Service (FRS) vehicles or MDUs [23,25]. The time taken for these resources to become operational is one of the significant drawbacks to wet decontamination. Dry decontamination (Figure 1), defined as ‘the topical application of absorptive materials to passively remove liquid contaminants from the skin surface’ [36] arose from the need for a more rapid, ‘ad-hoc’ intervention that could be performed by exposed persons themselves [37] before the arrival of specialist resources.

Dry decontamination using products such as Fullers Earth and bespoke absorbent material has previously been explored [38,39,40] however, these materials are not usually available in sufficient quantity for large-scale incidents. One of the major benefits of dry decontamination for mass casualty incidents is that it can be performed with any available absorbent material. Kassouf et al. [21] evaluated a range of absorbent materials for use as improvised dry decontaminants. Using simulants for sulphur mustard and soman (methyl salicylate (MeS) and diethyl malonate, respectively) and toxic industrial chemicals (parathion, phorate and potassium cyanide) the authors showed that absorbent materials readily available to ambulance and hospital staff, e.g., ‘blue roll’, were efficacious in removing liquid chemicals from ex-vivo pig skin. Dry decontamination had a higher efficacy compared to improvised wet decontamination (the improvised ‘rinse-wipe-rinse’ methodology) for all chemicals except for dried residues of potassium cyanide, suggesting that wet decontamination is still required for particulate contamination.

Amlôt et al. examined the efficacy of a novel improvised dry decontamination protocol in human volunteers contaminated with MeS [18]. ‘Blue roll’ and incontinence pads (two readily available absorbent materials on frontline response vehicles) were shown to be equally effective in removing MeS from volunteer’s forearms. The study suggested speed is more important than the choice of absorptive material. MeS recovery was consistently lower for all dry decontamination protocols compared to matched controls, indicating that any form of dry decontamination can reduce chemical contamination. However, blotting in combination with rubbing was shown to be the most effective dry decontamination protocol, underlying the importance of protocol design. Similar efficacy has also been demonstrated for non-ambulant casualties [34].

In a recent series of studies conducted by Southworth et al., the efficacy of dry decontamination was evaluated using MeS added to the shoulder, arm and leg of volunteers [41]. A significant decrease in MeS recovery was observed when participants conducted dry decontamination at 15 min post application when compared to a no decontamination control for the arm and the leg application sites. The shoulder application site however was not significantly different from control, highlighting that dry decontamination is effective at removing surface contamination but is less effective at harder to reach areas of the body. The study also found that the number of white-roll sheets used during dry decontamination was significantly associated with lower total recovery of MeS from the skin with on average three times less simulant recovered from volunteers that used 10 sheets or more compared to those that used less than 10 sheets [41]. Further studies are needed to determine whether time taken to decontaminate also correlates to a higher decontamination efficacy.

Dry decontamination has several advantages including rapidity, readily available decontamination media, and that exposed persons can self-decontaminate. Dry decontamination is now the default method of decontamination (following disrobe) for non-caustic chemicals in both UK and US federal IOR guidance [23,25]. Nonetheless, it is not considered the most appropriate method for all scenarios, for example corrosive chemicals, and consideration should always be given to using dry decontamination synergistically with wet decontamination (see section on multiple decontamination interventions in this review). Furthermore, a focus group study on the public perceptions of emergency decontamination [42] showed that a decontamination shower was perceived to be more effective than dry decontamination methods. Participants were mixed in their views as to whether they would comply with using blue roll to carry out initial decontamination. Additional work to improve the public understanding and acceptance of dry decontamination may therefore be needed.

### 2.4. There Are Optimised Parameters for Mass Decontamination Showering

Structured decontamination using MDUs (Figure 2) is an important provision of the specialist response to chemical incidents. These units typically have dedicated boiler systems to heat the water, a metered dosing system for introducing detergent for specific periods during the decontamination cycle and a means of containing contaminated water. A series of studies under the ORCHIDS projects [27] identified optimal parameters for mass decontamination showering resulting in the so-called ‘ORCHIDS protocol’ (Table 1). The effectiveness of this protocol has been demonstrated in several field trials [43]. The inclusion of active washing using cloths increased decontamination efficacy by up to 20% [19] and was particularly beneficial for children. Doubling of the showering duration from 3 min to 6 min showed no statistically significant improvement in decontamination efficacy. Chilcott et al. [26,44] suggested a maximum showering duration for the ‘ORCHIDS protocol’ of 90 s to partly offset the so-called ‘wash-in’ effect where skin penetration of chemicals can be temporally increased by washing [45]. However, as discussed later in this review, evidence to suggest a wash-in effect in humans is sparse and a re-evaluation of the evidence is required.

The use of detergents during structured mass decontamination, particularly to aid the removal of lipophilic substances has been suggested to remove up to 40% more contaminant than showering with water alone [29]. However, the data pertaining to this has not been published and as far as the authors are aware no human volunteer studies have been conducted to compare the efficacy of mass casualty decontamination with and without detergents. The importance of ‘active drying’ following wet decontamination was demonstrated where drying using towels could account for the removal of up to 50% of the contaminant during the decontamination cycle [26]. These parameters have been incorporated into both UK and US operational guidance [23,26].

### 2.5. Decontamination Methods Have Variable Efficacy for Liquid Contaminated Hair

The effectiveness of current decontamination protocols in removing contaminants from hair has received little attention. Compared to clothed skin, hair and scalp skin are relatively unprotected areas of the body and are likely to be significant sites of exposure during and after a chemical incident. Furthermore, there is potential for hair to bind chemical contaminants [46,47,48]. Whilst hair is a protective barrier for the scalp [28,49], certain chemicals diffuse rapidly through hair sebum to the follicles from which they can be absorbed [50,51]. In vitro studies of the efficacy of hair decontamination following exposure to VX [52] and the sulphur mustard simulants MeS and 2-chloroethyl ethyl sulphide (CEES) [53] revealed that showering alone was the least effective decontamination protocol, whereas the application of Fullers Earth or Reactive Skin Decontamination Lotion (RSDL) up to 45 min post exposure but prior to showering substantially improved decontamination efficacy but VX and MeS were persistent in hair post-decontamination (up to 27% and 57% of the contaminating doses, respectively). Subsequently, Spiandore et al. [54] demonstrated that MeS and CEES trapped in hair could rapidly desorb into the surrounding atmosphere with the amount of MeS in hair decreasing by a 2-fold factor in the first 2 h following exposure. After 24 h 0.12 ± 0.03 μg mg^−1^ MeS (8.6% of the initial dose) remained in the hair. These findings raise potential ongoing risks to the contaminated casualty, first responders and other members of the public.

Recent human volunteer field exercises and studies performed under more controlled conditions have shown mixed efficacy for the removal of chemical warfare agent simulants from hair. Chilcott et al. [28] demonstrated high decontamination efficacy for MeS contaminated hair during a field exercise in the US. However, these data should be treated with caution as the recovery of MeS in volunteers (including no-decontamination controls) across all application sites was low and highly variable. This could be due to methods of sampling (hair was not excised) or reflective of the challenges associated with using simulants under exercise conditions. Although the authors took steps to reduce confounds there were likely other factors (e.g., volunteer movements/interactions, fluctuations in weather conditions etc.) that are more difficult to control during a field exercise. A subsequent human trial performed under more controlled conditions (36) showed efficacy of Dry, Ladder Pipe System and Technical Decontamination methods at removing MeS from hair. Once again, hair was not removed in favour of swabbing the hair surface, and so the ability to detect the presence or absence of the simulant contaminant may have been confounded by the occlusion of the application site following decontamination. Furthermore, as acknowledged by the authors, the decontamination interventions performed in the study were conducted at unrealistically short timescales, not reflective of achievable intervention times during real operational response.

A series of human volunteer trials conducted by Collins et al. [55] using improvised dry/wet, interim wet, and mass decontamination showering methods at realistic timescales showed variable removal of two chemical warfare agent simulants, MeS and benzyl salicylate (BeS) from the hair and scalp of volunteers. Noting the limitations associated with the semi-quantitative sampling methods, dry decontamination was shown to reduce the amount of MeS and BeS remaining on the hair although results were only significant for the removal of BeS. Improvised wet decontamination (the rinse-wipe-rinse method) was shown to be more efficacious than dry decontamination but only for BeS. Data showing significant removal of BeS from hair is encouraging, particularly for a lipophilic simulant representative of a more persistent chemical threat such as Novichok [56,57] (based on predicted structures). The lack of significant reduction in MeS contamination for any decontamination intervention was postulated to be a factor of the greater lipophilic nature of the MeS and vegetable oil mixture used, with MeS binding to hair more effectively than BeS or alternatively the greater variability because of the increased loss of the more volatile MeS through vapourisation. In contrast to the data for hair, scalp swabbing showed that all the decontamination methods were highly effective at removing contamination from the scalp. These data are consistent with the high efficacy observed for the same decontamination methods used on human skin [58].

In summary, work investigating the optimal methods for the decontamination of hair has revealed mixed efficacy. Dry and wet decontamination appears to reduce the amount of contamination on hair and the scalp, but this is chemical-dependent. Reducing contamination via decontamination interventions would lower the ongoing exposure risk to the contaminated casualty (e.g., through off gassing), and the risk of secondary contamination for first responders, hospital receivers and other members of the public. Significant contamination remaining in the hair may require further action dependent upon toxicity of chemical and rate of off gassing. This could perhaps involve multiple rounds of decontamination or potentially the removal of hair, both of which would likely be a clinical decision informed by a dynamic risk assessment.

### 2.6. Multiple Decontamination Interventions Performed in Sequence Are More Efficacious That Single Interventions Alone

Following IOR, interim decontamination can bridge the gap between initial response and the arrival of technical resources as part of SOR. In the UK, while there is no prescribed procedure for conducting interim decontamination, the commonly recognised ladder-pipe system (two fire appliances with branches on ladders, creating a shower corridor through which casualties pass) is the most commonly exercised and accepted approach used by FRS responders. Most studies looking at decontamination interventions in sequence that include interim decontamination have also replicated this method, including a series of recent studies in which dry and wet IOR decontamination was conducted in sequence with interim and mass decontamination (SOR). These studies assessed the reduction in MeS on the skin of volunteers conducting these interventions in sequence. It was determined that combining dry and wet decontamination was more effective at removing the simulant from skin than conducting dry or wet decontamination alone [41], and when conducting interim decontamination following IOR, a trend towards decreasing simulant recovery was also observed [58]. In a follow-up study using a more persistent simulant, BeS, when SOR was introduced following IOR and interim decontamination, significantly less simulant was recovered compared to when not conducting interventions in sequence [59]. In a study of hair and skin decontamination [60], dry, interim and SOR decontamination conducted in sequence provided the highest reduction of hair and skin contamination. It also highlighted that conducting dry decontamination prior to specialist mass decontamination reduced possible secondary contamination through first responder contact with washcloths, towels and vapours within SOR units. These studies show that when combined, decontamination interventions can all contribute to potentially reducing morbidity and mortality. They also limit secondary contamination by reducing the build-up of high concentrations of chemicals in areas in which first responders are exposed. Emergency decontamination should therefore be a sequential process starting with less structured, improvised forms of intervention (i.e., IOR), through to structured technical decontamination (SOR), and that in no one intervention alone should be seen as sufficient.

### 2.7. Decontamination Must Be Casualty Focused to Facilitate Compliance

Studies that have examined casualty experiences and behaviour during decontamination [42,61,62,63,64,65,66,67,68] have suggested the way in which emergency responders manage an incident will affect the nature of the relationship between responders and members of the public, influencing how members of the public behave [67] and effecting the outcomes from the incident. Specifically, if emergency responders manage an incident effectively, members of the public will identify with them around a shared goal of undergoing decontamination [63,65,66]; this will result in increased public cooperation and compliance during the decontamination process [62,63,65]. It is therefore essential that emergency responders manage the incident to foster a shared identification between themselves and members of the public, to facilitate increased public compliance. Key actions that responders can take to achieve this shared identification revolve around effective communication and demonstrating respect for public needs. Responders should:(1)Communicate openly and honestly, providing regular updates about the nature of the incident and the actions that they are taking. Providing this information increases perceptions that responders are managing an incident in a fair and legitimate way and increases public willingness to comply with decontamination [42,63,65,66,68].(2)Communicate in a health-focused way about the need for decontamination. This includes explaining the nature of the threat (from contamination) and the efficacy of decontamination measures for reducing that threat [68]. This will ensure that members of the public understand the need for decontamination [61,63], and that they perceive the instruction to undertake decontamination as legitimate, thus increasing willingness to comply with decontamination instructions [42,62,66,67]. Specifically, responders should communicate: why decontamination is necessary, in terms of removing a contaminant from the skin and preventing any further risks to health; how decontamination will protect someone and their loved ones (e.g., reducing the risk of secondary contamination); and what decontamination will involve.(3)Casualties must also be able to effectively conduct decontamination. Evidence from decontamination exercises and field trials reveals that for both wet [62,63,66] and dry decontamination [18], provision of sufficient practical information is crucial for ensuring that casualties are able to effectively undertake decontamination, and that the process runs as smoothly and efficiently as possible.(4)To promote legitimacy of response, and hence increase casualties’ compliance, emergency responders must demonstrate respect for casualties’ needs. Demonstrating respect for casualties’ privacy and modesty will promote willingness to comply with decontamination [61,62,63,66]. While it may not always be possible to provide casualties with as much privacy as they would like, where this is not possible casualties’ concerns should be acknowledged, and the reasons for lack of privacy should be explained.(5)Respect members of vulnerable groups. There are four functional categories that may make someone more vulnerable during decontamination reduced ability to physically undertake decontamination; difficulty hearing or understanding decontamination instructions; social or cultural factors that make undergoing decontamination more difficult; existing health factors that may make them either more susceptible to the effects of the contaminant, or put them at increased risk while undergoing decontamination [69]. Responders must receive training and guidance for managing members of vulnerable groups and should put strategies in place to manage an incident involving these groups. As part of this process, responders should treat each individual as an expert in his or her own needs [69,70,71] demonstrating respect for casualties’ specific needs.

### 2.8. Decontamination Approaches Can Be Modified for Non-Ambulant Casualties and ‘Vulnerable Groups’

Casualties who are non-ambulant, either from pre-existing health conditions or as a consequence of chemical exposure, are less able to perform decontamination (Figure 3). A recent review [64] summarises the evidence for recommendations for supporting those with additional functional needs during decontamination. These include: allowing people to retain any functional aids that they may require; provide simple pictorial instructions; provide instructions in multiple languages; use body/sign language to communicate basic actions; and consider ethical, religious and cultural issues when asking people to disrobe such as ensuring that sufficient privacy is provided.

One key recommendation is the implementation of a buddy system, whereby a person with additional functional needs is supported through the decontamination process by an ambulant casualty [19,64,72,73]. A recent human volunteer trial demonstrated that less accessible parts of the body were more difficult to decontaminate [41] therefore, implementing a buddy system may not only assist those with additional functional needs, but also accelerate the decontamination process and efficiency for everyone [72]. Casualties will be willing, and even likely, to assist one another during emergencies [74,75], and a buddy system may represent a valuable resource during decontamination. To ensure that the help provided from one casualty to another is adaptive, it is essential that appropriate protective actions are communicated to casualties [76,77]. In this way, effective communication should be regarded as a key part of the approach for assisting casualties with additional functional needs.

Chilcott et al. [34] recently established some basic principles for non-ambulant casualty decontamination. Disrobe, and dry and wet decontamination protocols were optimised and evaluated in two human volunteer studies resulting in protocols that were rapid (3 and 4 min for dry and wet, respectively) and effective at removing contaminant (>95% reduction for most anatomical locations), whilst minimising skin surface spreading. Efforts should be made to further evaluate these protocols in combination and under more realistic conditions. Important considerations were raised by this study including the increased risk to first responders assisting with the decontamination. Cross-contamination of personal protective equipment (PPE) worn by the personnel decontaminating the non-ambulant casualties was observed on several occasions. Future studies should assess the risks to responders and determine appropriate safe practices for non-ambulatory casualty decontamination and PPE removal.

There is still some way to go in ensuring that recommendations for non-ambulant and vulnerable casualties are integrated into guidance and training for emergency responders. To address this, emergency responders should take part in regular exercises and training in mass decontamination, which should routinely include members of vulnerable groups.

## 3. Future Research Priorities

### 3.1. The Risks from Chemical Vapours

Ribordy et al. [78] reported persistent vapour concentrations of two volatile chemical warfare agent simulants, Purasolv ethyl lactate (PEL) and MeS in mobile wet decontamination units following the decontamination of exposed volunteers. Vapour concentrations occasionally exceeded exposure thresholds for most industrial chemicals, suggesting casualties and first responders could be at considerable risk from respiratory exposure. Since this study however, the introduction of disrobe and improvised decontamination as immediate actions will have largely mitigated this risk by reducing the amount of chemical contaminant remaining on casualties before they enter a MDU.

Work is still required to examine respiratory exposure to volatile chemicals within the context of improvised decontamination to determine if adjustments to response practices are required. Current UK guidance recommends the initiation of disrobe and improvised decontamination within 15 min of exposure [23]. Feldman [30] and Gaskin et al. [32] showed that certain items of clothing contaminated with MeS and chlorine gas off gassed within ten minutes of exposure. Furthermore, Spiandore et al. [54] showed a rapid initial rate of desorption of MeS from contaminated hair in vitro (tmax < 5 min). Respiratory exposure to volatile chemicals is likely to lead to systemic exposure considerably faster than dermal exposure. Therefore, certain volatile chemicals may cause incapacitation and disorientation to both casualties and first responders before the initiation of any form of decontamination, and certainly before the arrival of specialist decontamination capabilities. Consequently, time-delayed interventions such as the use of MDUs may have little effect on systemic levels of volatile substances. Immediate respiratory-based interventions (e.g., removal of contaminated clothes, relocation of exposed persons and the potential provision of vapour-protective masks) may need to be considered as an initial action in response to incidents involving volatile hazardous chemicals. Furthermore, for some volatile chemicals the need to disrobe or even decontaminate may be called in to question, especially in colder environments where the risks associated with the onset of hypothermia could introduce further challenges for the management of disrobed casualties.

### 3.2. The Choice of Chemical Simulant for Human Volunteer Studies

Until recently, with the exception of one study in which an oil-based simulant was used [79] all human volunteer decontamination studies, including those that have informed UK and US guidance used the simulants MeS or ethyl lactate (EL) [18,34,78,80,81,82,83]; volatile compounds (vapour pressures 4.6 Pa and 500 Pa at 25 °C) with similar physicochemical properties to sulphur mustard and sarin, respectively [84]. The range of chemical simulants safe to apply to humans is understandably more limited compared to those for in vitro studies [84] but James et al. [56] recently identified BeS as a simulant for more persistent, lipophilic chemicals such as Novichok [57] and subsequently demonstrated its suitability for evaluating decontamination efficacy in human trials [55,59,85].

Potentially toxic industrial chemicals (TICs) and chemical warfare agents span a wide range of physicochemical properties and may therefore show considerable variability in decontamination efficacies using current best practice. An important data gap concerns powders. As described, improvised decontamination methods have been proven to be effective at removing liquid contaminants from the skin but chemical agents in the form of powders (e.g., incapacitating agents such as fentanyl’s and several toxic pesticide products) represent a public threat that has received little attention. To our knowledge only one limited study has examined the efficacy of dry vs. wet decontamination methods for ‘particulate’ contamination in an in vitro model (16) and showed that dry decontamination was ineffective against particulate contamination, advocating the use of wet decontamination. This recommendation is reflected in US Federal Guidance (17) and from the authors experience is largely shared by the UK responder community. However, dry powders do not cross the skin barrier unless it is damaged. Penetration of powders through the skin is far more likely when in solution. Furthermore, the application of water to some powders can liberate toxic gases or cause more serious reactions. Therefore, Robust evidence is required to inform the optimal decontamination strategies for toxic powders.

### 3.3. Accurately Assessing Systemic Exposure

While the levels of simulant remaining on the surface of the individual is an important consideration, especially for corrosive/blistering agents and for assessing secondary contamination risk, the assumption that a reduction in external contamination levels translates to a reduction in systemic exposure has been the main outcome proposed in all studies to date. There is a possibility however that the amount of simulant removed during decontamination is not correlated to the amount that has or will become systemically available and therefore likely to induce systemic toxicity. To assess this, some studies have investigated the effects of decontamination interventions on urinary levels of applied simulants or their metabolites. The data obtained in these studies were confounded by high background levels from dietary or other sources [41,60]. To accurately assess systemic availability of simulants, an excreted simulant or simulant metabolite should be unique (i.e., should not be a common metabolite of other environmental exposures), non-endogenous (or at low enough baseline concentration), and controllable in study participants through dietary and consumer product use restrictions. A study conducted by James et al. [85] recently evaluated detection of parent MeS and BeS in the urine of volunteers, following skin exposure. This represents the first study to analyse un-metabolised simulants for the assessment of decontamination efficacy. Both simulants were present in high concentrations compared with baseline levels, with evidence to suggest that MeS becomes systemically available sooner than BeS. Peak MeS excretion was observed 60 min after dose whereas BeS reaching a peak excretion of 0.24% of the applied dose at between 12.5- and 21-h post-dose [85]. The identification of parent compound in urine is a major step towards not only identifying how effective decontamination is for removing chemicals from skin, but also to understand the effects upon systemic exposure to chemicals which can be used to model the symptoms casualties may experience and therefore the true efficacy of decontamination interventions. To take this into account, urine analysis during human volunteer trials should be used alongside typical skin and hair collection for a more holistic approach to understanding the efficacy of decontamination interventions.

### 3.4. The Significance of the ‘Wash-In’ Effect

First identified in limited in vivo experiments and later by in vitro studies [86,87,88,89,90,91], the enhanced dermal absorption of a small range of chemicals following wet skin decontamination has been reported. This so-called ‘wash-in’ effect [45] could have important implications for mass-casualty decontamination and has largely informed the adoption of dry decontamination as the primary default emergency decontamination protocol used by UK and US responders. Whilst there is a general lack of consensus on the mechanisms underlying the ‘wash-in’ effect, Moody et al. [45] have proposed several possibilities including the acidity/alkalinity of the soap, the influence of surfactant on skin barrier integrity, hydration of the stratum corneum and the effect of friction. Several in vitro studies support the ‘wash-in’ effect for chemicals applied to skin for 24 h prior to decontamination [86,87,88]. Fewer studies have investigated timescales (e.g., minutes) that are more relevant within the context of emergency decontamination [89,91].

While work has been conducted to identify the wash-in effect in vivo [92], clear evidence for a wash-in effect has so far proved elusive. This is possibly due to the need for more invasive sample collection (e.g., blood collection or catheterisation) to identify this temporal phenomenon. A lack of supportive results obtained from human volunteer studies utilising wet decontamination methods under real world conditions [45] however, suggest that the ‘wash-in’ effect may be an artefact of in vitro models.

Comparative studies have previously been conducted in which the same test chemical (VX) was applied both in vivo and in vitro [93,94]. respectively confirmed that decontamination protocols involving a soap water washing step significantly reduced the mortality of test animals exposed to a supra-lethal dose of VX, and that water washing of skin resulted in a significant reduction of penetrated VX through human skin. Thors et al. however demonstrated no significant reduction in penetration when washing was conducted 30 min post-dermal exposure, while Misik et al. determined that the ORCHIDS protocol was effective even after a delay of 1 h. This discrepancy is further complicated by in vivo evidence produced by Bjarnason et al. [95] in which soapy water decontamination was ineffective at reducing lethality of VX. There was no evidence however that this was attributed to the wash-in effect.

Given the potential ramifications of accelerating chemical skin adsorption through a ‘wash-in’ effect, controlled in vitro and volunteer studies utilising novel urinalysis methods for evaluating systemic exposure should be conducted. Studies using a wider range of chemicals to further elucidate the underlying mechanisms and to examine the importance, if any, of this phenomenon in the context of emergency decontamination.

### 3.5. Improved Communication Strategies

Whilst a significant body of research has developed in recent years applying crowd psychology to the development of optimised responder management and communication strategies in the context of CBRN incidents [69], further work is needed to understand how messages might be presented to promote understanding and adherence to IOR and SOR protocols. Providing clear, detailed instructions on how to complete dry decontamination has previously been shown to improve adherence to dry decontamination protocols (18). A recent novel development has been the use of an immersive virtual reality chemical incident scenario in a randomised controlled experiment, evaluating the impact of different messages emphasising the threat and severity of contamination and the effectiveness of IOR actions on willingness to follow responders’ instructions [68]. This study showed that messaging that emphasised both the potential threat of the contaminant, along with the efficacy of IOR protocols, resulted in greater reported willingness to disrobe and follow dry decontamination protocols. Through a novel use of technology to create a sense of realism, this study allowed the identification of different messaging strategies, which could be subject to further pragmatic testing in training and exercises. Whilst these preparedness activities can be argued to lack the characteristics of acute emergencies that might make crisis communication more challenging, they do represent important opportunities to understand the factors that might determine whether messaging is likely to be understood and accepted in real life scenarios. When combined with state-of-the-art understanding of public responses to real incidents and emergencies, these approaches are more likely to result in successful management of chemical incidents [96]. An additional challenge for public health communicators relates to how to communicate the need to perform decontamination protocols for slow-acting contaminants, or when symptoms of exposure are not immediately salient. For example, liquid sulphur mustard can have a latency of up to 12 h, necessitating an approach to casualty management and communication that may extend well beyond the initial scene and must involve receiving healthcare facilities and community settings. Further research could explicitly explore the relationship between responder management and public behaviour during real life incidents involving hazardous materials, including those involving wider community settings. Finally, in this review we have identified the importance of initiating emergency decontamination protocols quickly, and that this urgency can be compromised by the time taken for responders to arrive on scene to direct members of the public. One way to close the gap between exposure to a harmful chemical and the start of disrobe and decontamination is to improve public education such that improvised decontamination protocols might be initiated by the public, prior to the arrival of responders. A recent initiative in the UK, the “REMOVE” campaign has developed simple messaging about key actions that people should take to protect themselves and others during the initial response to an incident involving a hazardous substance. As part of the development of this campaign, focus groups and a survey with members of the public saw a positive appraisal of the materials and generated important insights into how the messages are understood and may be used [96,97]. Further work is needed to evaluate the impact of public education campaigns on preparedness for chemical incidents, including whether the provision of evidence-based messaging has impacted on decontamination performance and outcomes.

## 4. Conclusions

Timely and effective decontamination of exposed persons following a chemical incident can reduce harm and limit secondary contamination. Mass-casualty decontamination is resource intensive and complex. There is not a one-size-fits-all solution. Decontamination efficacy can be influenced by a multitude of factors including the number of exposed persons, the properties of the chemical agent, the speed of the response and the effectiveness of casualty and first responder communication and management.

Effective mass-casualty decontamination practices should be underpinned by robust scientific evidence. Findings from a series of in vitro and human volunteer studies have laid a strong foundation for current practices internationally but there is room for improvement. Several research gaps have been identified in this review including the risks from chemical vapours, the choice of chemical simulants for human studies, the need to more accurately assess systemic exposure and the wash-in effect and evidence-based communication strategies. These areas should be addressed as priorities to ensure that decontamination practices are efficacious, casualty-focused and adaptable to situational needs. Policy initiatives including the ‘REMOVE’ campaign in the UK and enhancing provision for mass casualty decontamination are underway to address some of these issues.

## Figures and Tables

**Figure 1 ijerph-18-03079-f001:**
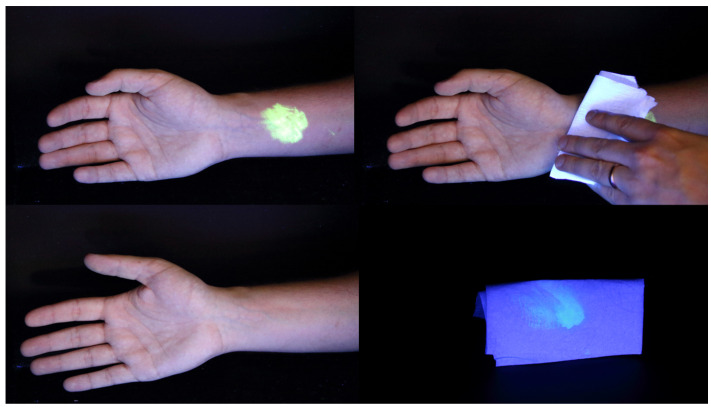
Representative images showing the removal of a fluorescent chemical simulant by dry decontamination using paper roll. Transfer of the simulant from the arm to the paper towel is evident (bottom right image). Post dry decontamination arm is shown in the bottom left.

**Figure 2 ijerph-18-03079-f002:**
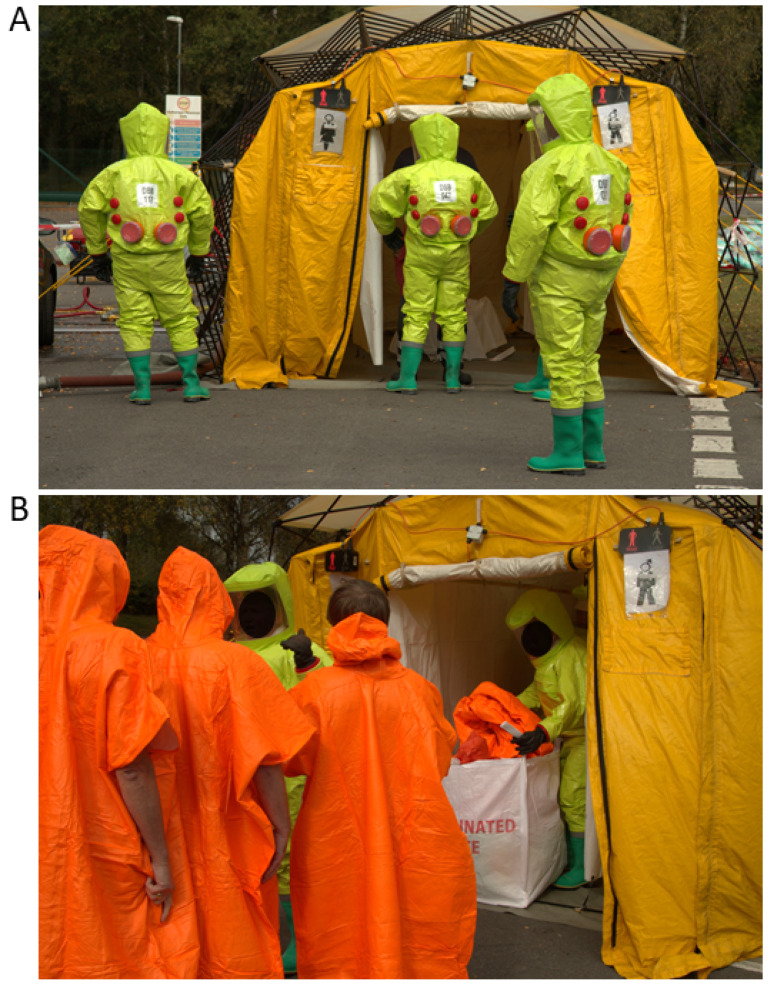
(**A**) A mobile mass decontamination unit (MDU) being set up by the emergency services during an exercise in the UK in 2019. (**B**) Disrobed casualties prepare to enter the MDU.

**Figure 3 ijerph-18-03079-f003:**
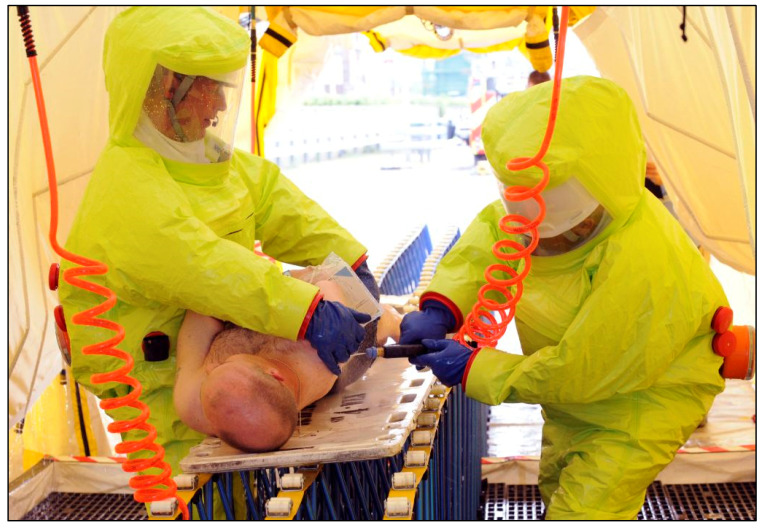
A non-ambulant casualty undergoing clinical decontamination during an emergency exercise, London, 2010.

**Table 1 ijerph-18-03079-t001:** The ‘ORCHIDS’ protocol for mass casualty decontamination using MDUs.

Parameter	Conditions
Temperature	35 °C
Duration	90 s
Active washing	Provision of cotton wash cloths
Detergent	0.5% detergent solution

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
