# Peer review of "Mass Casualty Decontamination for Chemical Incidents: Research Outcomes and Future Priorities"

_ijerph, 2021, doi:10.3390/ijerph18063079_

Round 1

Reviewer 1 Report

Review of Manuscript ID: ijerph-1118635 "Mass casualty decontamination for chemical incidents: research outcomes and future priorities"

The manuscript is a review of mass casualty decontamination following chemical incidents. The paper is pretty comprehensive and very well written, however, there are a few issues that needs to be addressed to fulfil publication. In the bullets below, all issues are separately described.

  1. The manuscript is a comprehensive review of decontamination research performed in the UK (and partly in the US by UK scientists), mainly by the authors of the present review in addition to Chilcott and colleagues. The recent studies are unique by using human volunteers and simulants as contaminants. However, the concept of rapid initiated decontamination has been proposed by others, e.g. Monteith and Pearce 2015, in terms of self-care decontamination. An overview of previous concepts and data supporting them would be beneficial for the manuscript to achieve a broader picture of the research field.
  2. The Fuller´s Earth and RSDL is mentioned in relation to separate studies in the manuscript, however, a discussion on specific decontamination products is completely lacking. The manuscript would benefit from a discussion the potential need for such products for certain contaminants, especially in a civilian perspective, associated to the ongoing research in this area.
  3. In row 322 and 499, the physicochemical properties of Novichoks is related to the use of BeS as a simulant. However, references is lacking in the manuscript to strengthen this correlation.
  4. This manuscript covers chemical incidents, however, general procedures are preferred to facilitate the management of rare events. In terms of HAZMAT/CBRN incidents, would the decontamination concept presented also be suitable following contamination of radioactive materials?
  5. In the paragraph covering the “wash in” effect, the authors suggests further studies using in vitro and human volunteer studies. By comparing previous work in vitro and in vivo using animal models including exposure to nerve agents and followed by soapy water decontamination (e.g. Misik et al 2012; Bjarnason et al 2008; Thors et al, 2020) correlations between decontamination efficacy in vivo and in vitro with symptom development in vivo may be performed. By including these comparisons and discuss the results, the paragraph on “wash-in” effects would be highly strengthened.   
  6. Minor issues:
    - The abbreviation BeS is used the first time on row 315, however, the full name benzyl salicylate is mentioned on row 357 and the abbreviation and full name is not initially connected. Please, go through the use of the full name and abbreviation in the manuscript.
    - Use the same spelling of mass casualty/mass-casualty throughout the manuscript.

Reviewer 2 Report

it is a nice piece that includes the impact of decontamination etc but I'm not sure the authors have provided anything original....is this not finished?? It reads more like a report some student in college might write... Take a look at the journals Conflict & Society, Conflict, Society and Development, and War and Society journals to see examples of what scholars wrote about.......But the bottom line is what is the theme they are reporting on and/or wishing to reveal that is new???

The manuscript does not include new experimental data or novel qualitative or quantitative analysis of the system's efficacy. Additionally, the concepts reported in my view are not sufficiently different from similar systems in other countries.

The paper addresses a difficult topic, which may not be suitable for the journal.  I have some reservations also on lack of evidence and argument put forward for the paper.  I do not recommend this paper to publish.   

Reviewer 3 Report

I thought this paper to to be very well done. Very complete, well referenced, well written, and extremely informative. 

My only suggestion is this.Only after reading most of the paper, does the reader understand that most of the paper involves wet contamination. The small section of dry contaminants later in the paper makes this clear. It would be helpful to the reader if near the beginning of the paper you make this clear. I spent a fair amount of time envisioning brushing off powders with blue pads.

Reviewer 4 Report

I consider this to be a very important contribution to the literature and I believe it definitely merits publication.

The authors are correct to note that our modern world poses serious risks of accidental chemical exposure, and their research is intended to bring the subject up to date with new findings from recent research.

The authors do a great job defining key concepts and they are careful to explain what can happen to people who are exposed to toxic chemicals.  

The paper is well-organized and easy to read in spite of the fact that some of the information is (to be expected) highly technical.

The findings regarding speed of operations, disrobing, wet and dry decontamination, parameters for showering, hair, multiple decontaminations, human behavior, non-ambulant victims, and communication are vital for today's first responders and specialized teams.

The authors also describe future research priorities, including a better understanding of vapors, wash-in effect, and human volunteer and virtual reality studies.

Because the paper is well-written and thorough, I really do not have any significant recommendations.  

One suggestion is that the authors could mention that chemical exposure could result from terrorism (besides the other sources mentioned).  Another is to explain what types of questions could be asked to better understand crowd psychology.  Finally, the paper could have one last sentence to summarize the goals of decontamination for the future.

Overall, very nicely done!  I comment the authors and strongly recommend publication.

Round 2

Reviewer 1 Report

Review of Manuscript ID: ijerph-1118635

The manuscript has only a few minor concerns (listed below) to fulfill publication.

Minor issues

  1. Row 405: only the abbreviation BeS should be used.
  2. For the Novichoks, it should be noted that the physicochemical properties are predicted in the selected reference.

Author Response

Thank you for the continued review of the manuscript. We have corrected the highlighted points as follows:

1. Row 405: only the abbreviation BeS should be used.

Response: Corrected

2. For the Novichoks, it should be noted that the physicochemical properties are predicted in the selected reference.

Response: Corrected. Line 323 to 325 'Data showing significant removal of BeS from hair is encouraging, particularly for a lipophilic simulant representative of a more persistent chemical threat such as Novichok (based on predicted structures) [56,57]'.

Reviewer 2 Report

Dear authors

The quality of the report continues to be low and I do not considered the manuscript suitable for publication

You stated this is a narrative review but beside conclusion I could not find this information

The study is limited by use of only self-report measures, by poor writing that makes it hard for me to fully grasp the constructs of interest included in the analyses, and by primarily replicating existing findings (which is important in science) rather than adding new information to our knowledge base. 

Note that I am a well-published author and physcian with 30 years of expierience in emergency medicine so I dont think my review is unbalance.
This is just my opinion you dont have to agree with it.
Wish you all the best.

Author Response

We thank the reviewer for taking the time to review the revised manuscript. However, we continue to disagree with the reviewer's negative stance on the manuscript. The review takes a negative view but the reviewer does not provide any examples from the paper to support their position. As a result, we have no opportunity to revise or refute any claims made by the reviewer. In that respect, the reviewer's response doesn't really constitute a peer review but rather just a statement of personal opinion. 

We continue to disagree with reviewer two and but reassured that the other three reviewers were highly supportive of the manuscript.